# Syntheses and Characterizations of CuIn_1-x_Zn_x_Se_2_ Chalcopyrite Nanoparticles

**DOI:** 10.3390/ma15041436

**Published:** 2022-02-15

**Authors:** Khedidja Benameur, Younes Mouchaal, Kheireddine Benchouk, Abdelkader Laafer, Regis Barille

**Affiliations:** 1Laboratory of Physics of Thin Films and Materials for Electronics (LPCMME), University of Oran 1 Ahmed BenBella, BP 1524 Oran El-Mnaour, Oran 31000, Algeria; benameurkhedidja@gmail.com (K.B.); mouchaal.younes@gmail.com (Y.M.); kbenchouk08@gmail.com (K.B.); 2Faculty of Exact Sciences, University of Mascara Mustapha Stambouli, B.P. 305, Mascara 29000, Algeria; 3Laboratory of LSTM, Department of Renewable Energy, Faculty of Technology, University of Blida 1, Soumaa Street No.270, Blida 09130, Algeria; a.laafer@gmail.com; 4MOLTECH-Anjou, University of Angers/CNRS (UMR6200), 49045 Angers, France

**Keywords:** CuIn_1-x_Zn_x_Se_2_, nanoparticles, chalcopyrite, solvothermal

## Abstract

CuIn_1-x_Zn_x_Se_2_ powders with various atomic percentages (x = 0, 0.05, 0.11, 0.16 and 0.21) were synthesized with the solvothermal method using metal chlorides and ethylendiamine as sources of precursors and a solvent, respectively. The experiment aims to investigate the effect of atomic percentages of Zn_x_ compounds on the structural and optical properties of CuIn_1-x_Zn_x_Se_2_ in order to improve future technological applications based on this material. The powders’ chalcopyrite phases were identified by X-ray diffraction. Energy dispersive X-ray spectroscopy analysis revealed the presence of Cu, In, Zn and Se with the expected atomic ratio of Zn/(In + Zn). Scanning electron microscopy and transmission electron microscopy analysis showed that the powders have large-scale desert rose-like structures. The nanopowders’ optical study by UV-visible spectrophotometry showed that the CuIn_1-x_Zn_x_Se_2_ energy gap values increase with the molar fraction of Zn_x_. A change from 1.15 to 1.4 eV was observed.

## 1. Introduction

Research on solar energy materials is an important issue for the maximum conversion of sunlight on Earth. A particular material relevant for solar energy harvesting is the I–III–VI_2_ group of compounds, where the group I metal is typically Cu, the group III metal is typically In or Ga, and the group VI element is generally Se or S [1]. These semiconductor compounds are important materials with wide applications due to their satisfactory thermal stability and high absorption coefficients, typically in the order of 10^5^ cm^−1^ [2,3]. However, despite the interesting properties of CuInSe, the presence of indium is a drawback as it is a scarce material on Earth, making it an obstacle to the economic development of devices based on these materials. To overcome this problem, studies have proposed to partially replace indium with other cheaper and better abundant elements on Earth such as Ga, Al and Zn. This will help to adjust the energy gap by adapting the Zn, Ga, Al…/In atomic ratio, which optimizes the absorption of the solar spectrum. It is also assumed that transition metals (TM) such as Zn have a technological advantage for their favorable magnetic, optical and electronic properties required for spintronic materials and optoelectronic devices. A very promising approach is defect engineering through doping. With different physicochemical properties, the dopants can effectively destabilize the parent oxide host and significantly enhance ionic diffusions or transportations by inducing local structural distortions, stress/strain as well as ionic defects.

In order to synthesize these chalcopyrite compounds, there are several techniques, either vacuum-free techniques such as spray pyrolysis [4], spin coating [5], dip coating [6] and solvothermal synthesis [7,8], or vacuum techniques such as co-evaporation [9]. The latter technique requires stringent conditions for high-vacuum synthesis, which results in higher production costs and higher energy consumption. To avoid or reduce all these drawbacks, we used the solvothermal method here. It is a simple and fast technique without high-vacuum conditions and with low-cost equipment. The principle is based on chemical reactions between the precursors and the solvent in autoclaves sealed at a higher pressure and at a temperature higher than the boiling of the solvent, which allows several chalcopyrite materials to be synthesized [10]. In this work, the development of CuIn_1-x_Zn_x_Se_2_ or CuInZnSe using the solvothermal method is concerned in order to adapt the lattice parameters and the direct gap energy of CuIn_1-x_Zn_x_Se_2_ powders [2]. Their crystalline structures and their optical properties were characterized by X-ray diffraction (XRD), energy dispersive X-ray spectroscope (EDS), scanning electron microscopy (SEM), transmission electron microscopy (TEM) and UV-visible absorbance in order to explain how the addition of Zn influences the microstructure and the composition of the absorbing powders of CuInSe_2_.

The present study contributes to improve the knowledge of these new materials to compete with current technologies for most developing thin films.

## 2. Experimental Details

### Synthesis of CuIn_1-x_Zn_x_Se_2_ Nanopowders

CuCl (99.999% Sigma-Aldrich), InCl_3_ (99.999%, Sigma-Aldrich, St. Louis, MO, USA), ZnCl_2_ (99.999%, Sigma-Aldrich) and Se (powder 99.5%, Sigma-Aldrich) were used in the form of powders as precursors. Ethylenediamine (98%, BIOCHEM Chemopharma, Cosne-Cours-sur-Loire, France) was used as a solvent.

The material synthesis was executed with the following method in a glove box under argon atmosphere. First, 0.12 g of elemental Se was dissolved in 20 mL of ethylenediamine with a magnetic stirring for 2 h at room temperature. Then, we added 0.08 g CuCl, 0.1 g InCl_3_ and 0.01 g ZnCl_2_. The mixture was be stirred continuously for 2 h. Then, the mixture was loaded in a stainless steel autoclave with a capacity of 25 mL and covered with Teflon. The system was maintained at a temperature of 180 °C for 24 h in an electric oven. After this reaction time, the autoclave was cooled down naturally at room temperature. Using a centrifuge, the powders were rinsed several times with distilled water and ethanol to ensure reagent-free particles (Figure 1). Finally, the product was dried under vacuum at 80 °C for 6 h to provide black powders. For each experiment, we changed the InCl_3_/ZnCl_3_ ratio in order to study the effect of x (Zn/(In + Zn)) molar fraction on the physical properties of CuIn_1-x_Zn_x_Se_2_ nanopowders. The stoichiometry quantities of In and Zn were systematically adjusted according to the value of x by adjusting the proportion of reactants (0.1 to 0.17 g of InCl_3_ and 0.01 to 0.04 g ZnCl_2_) in the synthesis, and the same amounts of CuCl and Se used in the previous synthesis were used during the synthesis. The proposed mechanism involved in this solvothermal reaction for the growth of CuInZnSe nanoparticles can be summarized by the following equation:

Ethylenediamine
CuCl + xInCl3 + (1 − x)ZnCl2 + 2Se  Cu(InxZn1-x)Se2 nanocrystals + byproducts 180 °C

## 3. Characterization

An X-ray diffraction instrument (XRD, l = 1.5406 Å) equipped with graphite monochromator Cu Ka radiation (Bruker D8 Advance, Billanca, Massachusetts, USA) was used to examine the crystal structure of the as-synthesized product, and the scanning rate was 0.01 deg/s. The operation voltage and current were maintained at 40 kV and 40 mA, respectively. The morphology of the as-synthesized powders was analyzed by a field-emission scanning electron microscope (SEM, INCA X Max, Oxford, England, accelerating voltage of 15 kV) with an energy dispersive X-ray spectroscopy (EDS) attachment, and transmission electron microscopy (TEM) analyses were carried out using JEOL JEM (1400—120 Kev, Tokyo, Japan). The FT-IR spectrometer (BRUKER Vertex 70, Billanca, Massachusetts, USA) with a spectral range of 450−4000 cm^−1^ was used to assess the molecular structure of the specimens. The optical absorption spectra were measured by a UV-vis double-beam spectrophotometer (RAYLEIGH UV-2601, Trafford Wharf Rd, Trafford Park, Stretford, Manchester, UK). Concerning XPS measurements, an Axis Nova instrument from Kratos Analytical (Trafford Wharf Rd, Trafford Park, Stretford, Manchester, UK) spectrometer with an Al Kα line (1486.6 eV) as an excitation source was used. The core level spectra were acquired with an energy step of 0.1 eV and using a constant pass energy mode of 20 eV (energy resolution of 0.48 eV). Concerning the calibration, binding energy for the C1 hydrocarbon peak was set at 285 eV. Then, data were analyzed with the CasaXPS software (Casa Software Ltd., Bay House, 5 Grosvenor Terrace, Teignmouth, UK). All the measurements were carried out at room temperature.

## 4. Results and Discussion

### 4.1. Structural Properties of CuIn_1-x_Zn_X_Se_2_ Nanopowders

The crystallinity and the phase of CuIn_1-x_Zn_x_Se_2_ powders with various values of x were analyzed by XRD analysis. The X-ray diagrams of the CuInZnSe powders presented in Figure 2 show that all the powders crystallized with a tetragonal chalcopyrite structure, confirmed by the presence of well-defined peaks at (112), (220), (312), (400), (332) and (228) and low-intensity peaks at (101) and (103). Another characteristic that can be obtained from the diffraction data is the decrease in the intensity of the diffraction peaks when the molar fraction of Zn_x_ is increased [11]. This observed result indicates a better crystallinity for low Zn_x_ atomic component in the CuInZnSe nanopowders. In addition, we noticed that the diffraction peaks for CuInZnSe move towards slightly higher diffraction angles (Figure 3) due to the increase in the atomic ratio Zn/In.

The incorporation of Zn in the CuInSe_2_ structure caused a decrease in the volume of the elementary lattices (Table 1) because the radius of the Zn^2+^ ion (radius = 0.74 Å) is smaller than that of In^3+^ (radius = 0.81 Å). These results are in agreement with previous results in literature, where it was observed [12] that CuInZnSe peaks were shifted to higher diffraction angles with the incorporation of Zn.

The lattice parameters a and c of the CuIn_1-x_Zn_x_Se_2_ powders were calculated with Bragg’s law, which gives the connection between the interplanar spacing ($d_{hkl}$) of the reflection set of crystalline planes and the sinus of half of the Bragg’s angle (2θ) that the diffracted beam makes with the transmitted beam from the sample. Moreover, $2d_{hkl}\sin\left(\theta_{hkl}\right)=n\lambda \;d_{hkl}\;$has well-defined mathematical connections for a given class of crystal structure. $d_{hkl}$ is linked to the lattice parameters through the *h*, *k* and *l* Miller indices and is given by the reticular distance of the tetragonal structure of two successive lines:$$\frac{1}{d^{2}}=\frac{h^{2}+k^{2}}{a^{2}}+\frac{l^{2}}{c^{2}}$$

The mean crystallite size of CuIn_1-x_Zn_x_Se_2_ powders with various values of x was determined using the Debye–Scherrer formula:$$D=\frac{K\lambda }{\beta \cos\left(\theta \right)}$$
where λ is the wavelength of the X-rays, β is defined at the full width at half maximum value (FWHM) and θ is the Bragg diffraction angle.

*K* is a constant, and it depends largely upon the crystallite shape. Values of 0.7 to 1.70 have been assumed for *K* by different investigators. Since in most cases the crystallite shape is unknown and may vary from crystal to crystal, it is probably best to define D as the mean dimension of the crystallite perpendicular to the diffracting plane (Dhkl). This definition has been shown to give a *K* value of about 0.9 when β is taken as the half-maximum line breadth. Therefore, we use this well-accepted value in the present work [13].

We note that in Figure 4, showing the variation in the grain size of the CuInZnSe powders as a function of the atomic ratio Zn/(Zn + In), the grain size or the crystalline quality first increases, indicating that the zinc atoms occupy vacant indium size. Then, the grain size decreases at the threshold of 0.05 with the increase in the atomic ratio Zn/(Zn + In). The zinc atoms are substituted to indium atoms. The decrease in the crystallite size is probably due to the small ion size of Zn^2+^ (radius = 0.74 Å) compared to that of In^3+^ (radius = 0.81 Å). These results are similar to those obtained with CuInGaSe powders [11] and radiofrequency (RF)-sputtered CuInZnS thin films [14], where the incorporation of impurities as Zn or Ga in the CuInSe_2_ compound reduced the intensity of the peaks and the grain sizes from 24.8 to 22.5 nm. The behavior of Zn is the same as the behavior of Ga in CuInSe powders.

The dislocation density (δ) may also be considered as a measure of the defects amount in the compound. δ is defined as the length of the dislocation lines per unit surface of unit cell of the crystal and was calculated using the Williamson and Smallman formula [15]:δ = n/D^2^

where n is a factor that gives the minimum dislocation density when n has a value of 1, and D is the crystallite size. It has become a fundamental building block in the explanation of dislocations and incorporation of Zn into crystal structures.

Figure 4 shows the variation in the dislocation density as a function of the atomic ratio Zn/(In + Zn). As shown in Figure 4, the dislocation density first decreases with small variations of Zn/(In + Zn) until the atomic ratio equals 0.11. Then, δ increases with higher values of the atomic ratio Zn/(In + Zn), which shows that more defects are created with the penetration of Zn in CuIn_1-x_Zn_x_Se_2_ powders [16,17]. Figure 4 shows that the best crystalline structure is achieved only up to an atomic ratio Zn/(In +Zn) of 0.05.

### 4.2. Morphology and Compositional Analysis of CuIn_1-x_Zn_x_Se_2_ Nanopowders

EDX was used to determine the chemical composition of the elements found in CuIn_1-x_Zn_x_Se_2_ powders. The EDX analysis leads to a large uncertainty regarding the measurement of powders because the analyzed medium is inhomogeneous since the analysis concerns the grains of the powder and, possibly, some empty zones between the grains. To gain precision, we analyzed several points for each sample, which decreased the uncertainty of the measurement. Systematically, we measured ten zones per sample, and we gave the average value in Table 2. The EDX results presented in Table 2 reveal the presence of Cu, In, Zn and Se with the adequate atomic ratio of Zn/(In + Zn). The EDX results presented in Table 2 show that the atomic percentage of indium decreased with the increase in the concentration of Zn_x_ in CuIn_1-x_Zn_x_Se_2_ powders. The apparent reduction of indium in CuIn_1-x_Zn_x_Se_2_ powders in the form of Zn_x_ is due to an anti-site substitution of Zn_x_ in the sites of indium within the chalcopyrite structure [14]. A slight variation was also noticed between the composition of the individual particles measured by EDX and the ratio of the precursor taken at the starting precursor composition. If we take into account the uncertainty, there is no significative difference. Moreover, the measurement gives a trend that, as expected, the at % Zn of the final product increases with the initial concentration.

The morphologies of CuIn_1-x_Zn_x_Se_2_ powders with various of values of x, synthesized by the solvothermal method, were observed by SEM and TEM. The SEM images in Figure 5 show that the CuIn_1-x_Zn_x_Se_2_ powders include ball microspheres and interconnected nano-platelets. These nano-objects resemble desert rose-like structures at a higher magnification [18,19], as shown in Figure 6. The size of the nanoballs is larger than the average crystallite size determined by XRD. This can be explained by the fact that each particle observed by SEM is formed of several crystallized grains. Additionally, we do not measure the same entity; XRD gives the size of the grains perpendicularly, while the SEM gives the size of the grains on the surface, so parallel to the surface. These results show that the particle size decreases with the percentage increase in Zn in CuInZnSe powders.

TEM images are used to obtain more information on the structure and shape of the nanoparticles that make the CuIn_1-x_Zn_x_Se_2_ powders with various contents of Zn_x_ synthesized by the solvothermal method. The TEM images in Figure 7 show that the powders consist of spherical-shaped agglomerated nanoparticles. The morphology of the particles not perfectly spherical also contributed to their deviation from a mean size shape in which many small building blocks are connected in a large network. The size of these particles was determined by statistical counting using the ImageJ software, ranging from 55 to 25 nm. These results suggest that they give a correct idea since they are confirmed by the DRX. The average size of the nanoparticles as determined by TEM indicates that the particles are likely not small enough to exhibit size quantization effects, although nanoparticles are not truly “quantum dots”. The material is largely aggregated; however, we are optimistic that these precursors may yet provide a route to colloidal chalcopyrite quantum dots [20]. On the other hand, these compounds can be used for the preparation of CuInSe_2_/NiFe_2_O_4_ nanocomposites by a simple MOF-assisted synthesis at moderate temperatures from the perspective of photocatalyst applications in the degradation of endocrine disruptors in an aqueous medium [21].

### 4.3. Elemental Analysis of CuIn_1-x_Zn_x_Se_2_ Nanopowders

In Figure 8, the shape of the spectra of the different elements, Cu2p_3/2_, In3d_5/2_, Zn2p_3/2_ and Se_3d_ present in the samples show that they are not oxidized. This is particularly explicit in the case of Cu; it is known that the spectrum of copper oxide is completely different from that of Cu present in chalcopyrite compounds (inset Figure 8a) [22,23]. The binding energies of the different elements can be deduced from Figure 8. For Cu 2p_3/2_ and 2p_1/2_, they are 932.4 eV and 952.2 eV, respectively. From Figure 8b, the In3d_5/2_ and In 3d_3/2_ binding energies are 444.9 eV and 452.4 eV, while the values estimated for Se3d_5/2_ and Se3d_3/2_ are 54.5 eV and 55.7 eV, respectively. These binding energies are in agreement with those reported in the literature for chalcopyrite structures [24,25]. Figure 8c shows the XPS spectrum of Zn2p, where it can be seen that Zn2p_3/2_ and Zn2p_1/2_ are situated at 1021.9 eV and 1044.9 eV, respectively. This means that there is a small increase in the Zn orbital energies in comparison with the expected values for pure Zn [22]. This small shift corresponds to electron exchange with Se, which confirms the Zn substitution to In.

In Figure 9, it can be seen clearly that the intensity of the Zn signal increases with the x value. The x values were determined by EDS, and it is remarkable that the relative intensity of the lines detected by XPS follows these values fairly closely. Thus, the XPS study confirms the presence of Zn and its interaction with the atoms of the CuIn_1-x_Zn_x_Se_2_ compound.

### 4.4. Optical Properties of CuIn_1-x_Zn_x_Se_2_ Nanopowders

FT-IR spectroscopy is known as one of the most convenient methods to find information about the chemical bonding of matter and elemental constituents [26]. The infrared spectra in Figure 10 were obtained after the analysis of CuIn_1-x_Zn_x_Se_2_ powders with infrared light. These spectra show that the absorption peaks attributed to ethylene diamine are absent. Because of the modest amount of Zn used, the spectra are largely similar. The peaks typical of CuIn_1-x_Zn_x_Se_2_ bounds decrease in intensity as x increases. The typical peaks at 482, 561, 651, 671 and 970 cm^−1^ [27] represent Zn-Se vibrations. Peaks representing Zn-Se vibrations may be found at 482, 717 and 823 cm^−1^. This confirms the effective formation of CuIn_1-x_Zn_x_Se_2_ composites. Another feature that can be obtained from IR data is that the absorption of CuInZnSe powders shifts to shorter wavelengths with increasing atomic ratio Zn/(In + Zn) [28]. These FT-IR results confirm that ethylenediamine can behave as a complex ligand and form stable complexes with CuCl, InCl_3_, ZnCl_3_ and powdered Se.

Absorbance measurements on CuIn_1-x_Zn_x_Se_2_ powders using UV-vis spectroscopy in the wavelength range from 200 to 1100 nm at room temperature were performed to deter-mine the optical parameters of CuIn_1-x_Zn_x_Se_2_. The sample was dispersed in absolute ethanol (≥99.9%) under intense sonication for 20 min and ethanol was used as a reference. We calculated the band gap of CuIn_1-x_Zn_x_Se_2_ from UV-vis absorption data using the Tauc Plot method.

In the Tauc Plot method, it is necessary to extrapolate to zero the linear part of the curve representing the Davis and Mott relation (αhν)^n^ = K(hν − Eg) [29],where α is the absorption coefficient, K is a constant, hν is the photonic energy, Eg is the energy of the band gap and n represents the nature of transition. For direct band gap material, n = 2, while for indirect band gap material, n = 1/2. The Davis and Mott relation can be further expressed as (2.303 × A × 1240/λ)^n^ = K(1240/λ − Eg) [30], where A and λ are the absorbance and wavelength, respectively, obtained from the absorption spectra of the nanoparticles. A plot of this relation gives an absorption curve in which its tangent gives the energy band gap of the nanoparticles [30].

Chalcopyrite CuIn_1-x_Zn_x_Se_2_ is a direct band gap semiconductor, so to determine their band gap, we extrapolate the linear part of the curve representing the (αhv)^2^ function to zero. Figure 11 shows the result of (αhν)² as a function of hν. Based on these results, the energy of the band gap can be estimated with 0.05 eV error. Figure 12 represents the variation in the values of the forbidden band as a function of the atomic ratio Zn/(In + Zn). From these results, we deduce that the increase in the atomic ratio Zn/(In + Zn) is followed by the increase in the band gap, and this may be due to the displacement of the CuIn_1-x_Zn_x_Se_2_ conduction band to higher energy levels and the valence band to lower energy levels. Therefore, the band gap of these compounds increases [31]. All these results show that the role of zinc is identical to that of aluminum in Cu (In, Al) Se_2_ powders [32,33] as well as gallium in Cu (In, Ga) Se_2_ powders [11,31,34,35]. In the latter case, Zn is cheaper than Ga concerning the ease of acquisition because it is a natural component of the Earth’s crust and is an integral part of our environment, while Ga is a rare chemical element. In nature, gallium is present only to a small extent and mainly as a mixture in aluminum, zinc or germanium ores; gallium minerals are very rare.

## 5. Conclusions

In this work, we synthesized CuIn_1-x_Zn_x_Se_2_ nanoparticles using a simple, less expensive and faster method, the solvothermal method. The effects of various Zn contents on the properties of the as-synthesized CuInZnSe nanoparticles were investigated in detail.

The results of CuInZnSe synthesis show that the structural and optical properties of CuIn_1-x_Zn_x_Se_2_ are influenced by the atomic percentage of Zn. XRD diagrams have shown that the chalcopyrite phase is well formed when the atomic percentage of zinc is lower in CuIn_1-x_Zn_x_Se_2_. Through SEM and TEM analyses, CuIn_1-x_Zn_x_Se_2_ powders have also been observed to be composed of agglomerated nanoparticles of spherical shape, resembling desert rose-like structures at higher magnification. Optical measurements indicate that CuIn_1-x_Zn_x_Se_2_ has a direct band gap which is increased with the incorporation of zinc into CuIn_1-x_Zn_x_Se_2_. The results obtained in this study demonstrate that Zn can be an effective candidate for substitution in chalcopyrite structures.

Therefore, the as-synthesized CuIn_1-x_Zn_x_Se_2_ nanoparticles can be used for the preparation of CuInSe_2_/NiFe_2_O_4_ nanocomposites by simple MOF-assisted synthesis at moderate temperatures from the perspective of photocatalyst applications in the degradation of endocrine disruptors in an aqueous medium.

## Figures and Tables

**Figure 1 materials-15-01436-f001:**
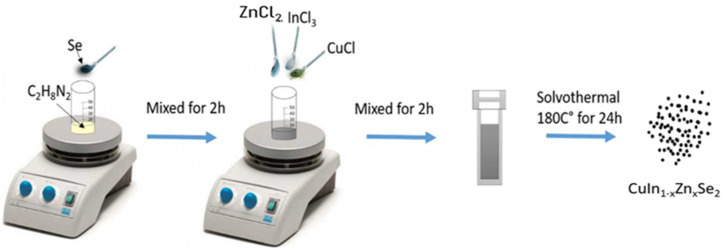
Solvothermal synthesis of CuIn_1-x_Zn_x_Se_2_ powders.

**Figure 2 materials-15-01436-f002:**
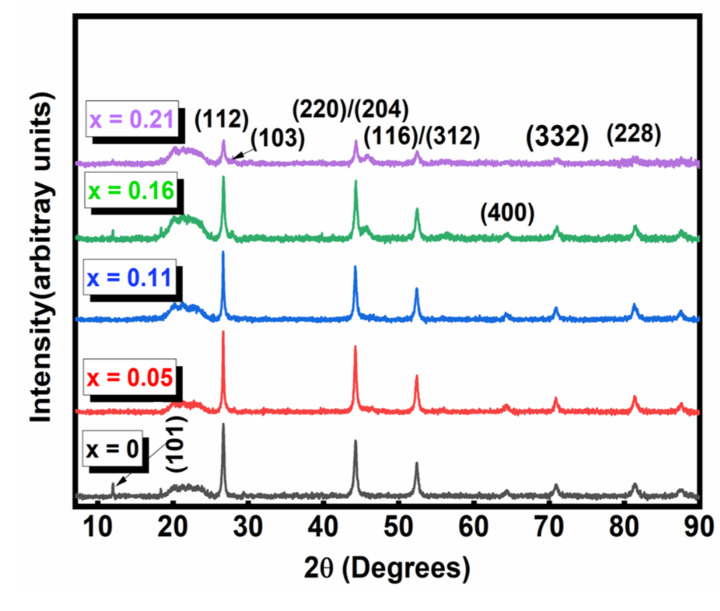
XRD patterns for CuIn_1-x_Zn_x_Se_2_ powders.

**Figure 3 materials-15-01436-f003:**
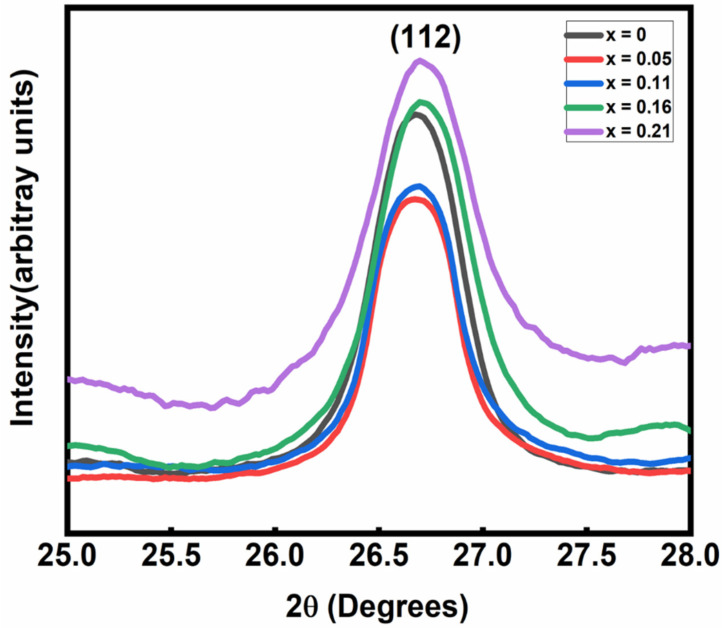
Zoom of XRD patterns for CuIn_1-x_Zn_x_Se_2_ powders.

**Figure 4 materials-15-01436-f004:**
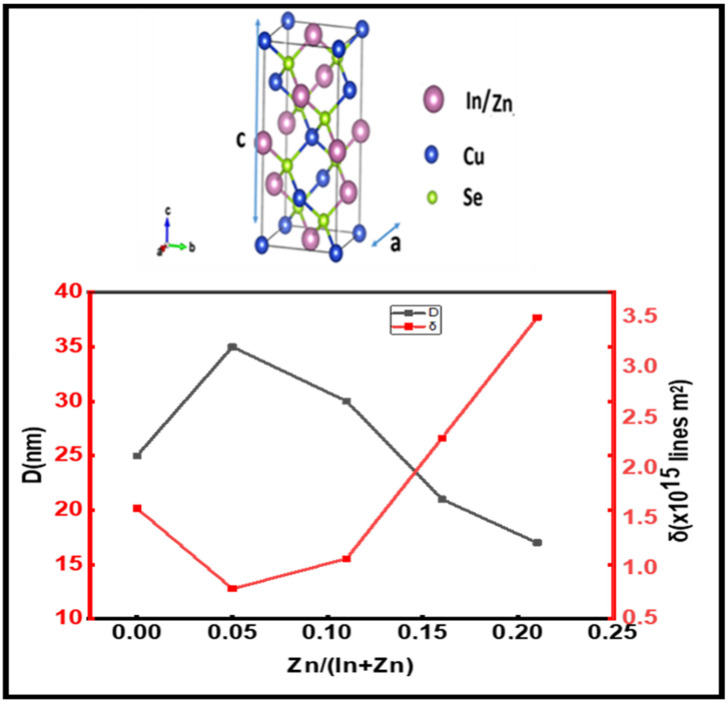
Variation in the dislocation density (δ) and grain sizes (D) for CuIn1_-x_Zn_x_Se_2_ powders as a function of the atomic ratio Zn/(In+Zn).

**Figure 5 materials-15-01436-f005:**
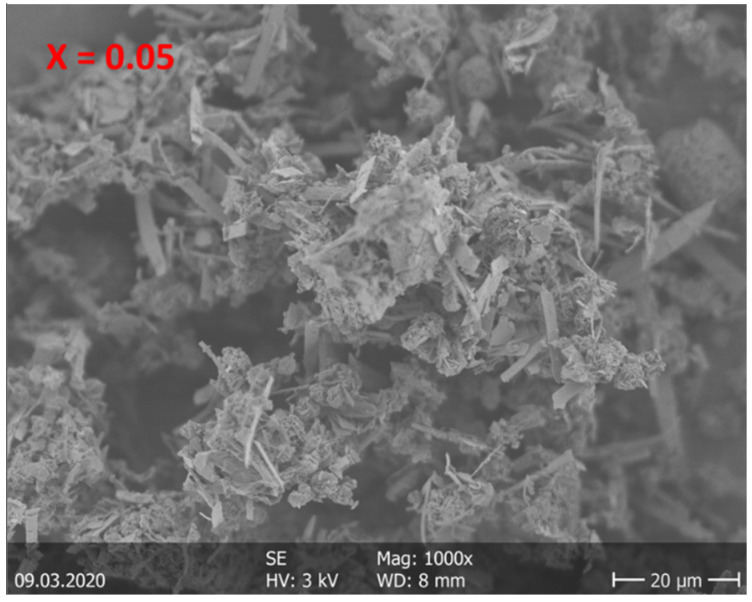
SEM images of CuIn_1-x_Zn_x_Se_2_ powders.

**Figure 6 materials-15-01436-f006:**
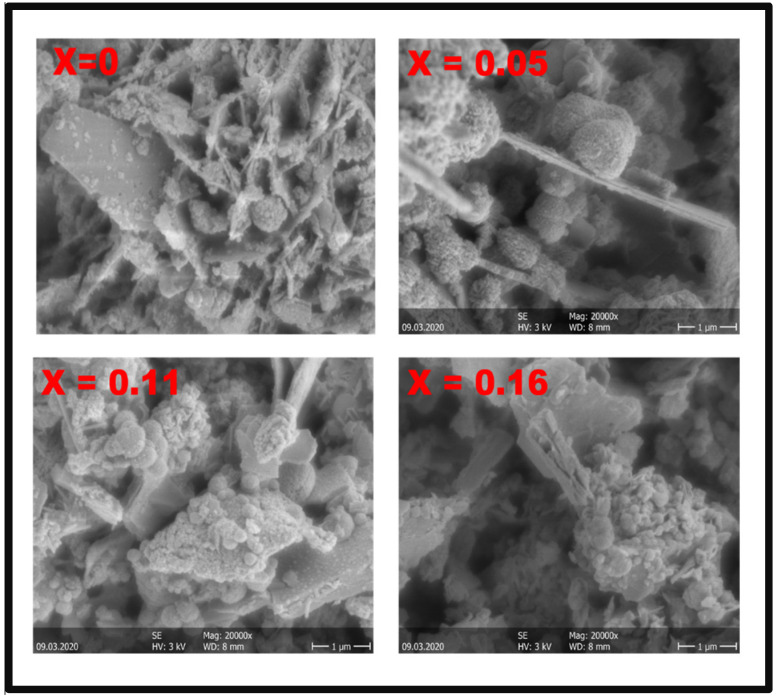
Zoom on CuIn_1-x_Zn_x_Se_2_ powders with SEM.

**Figure 7 materials-15-01436-f007:**
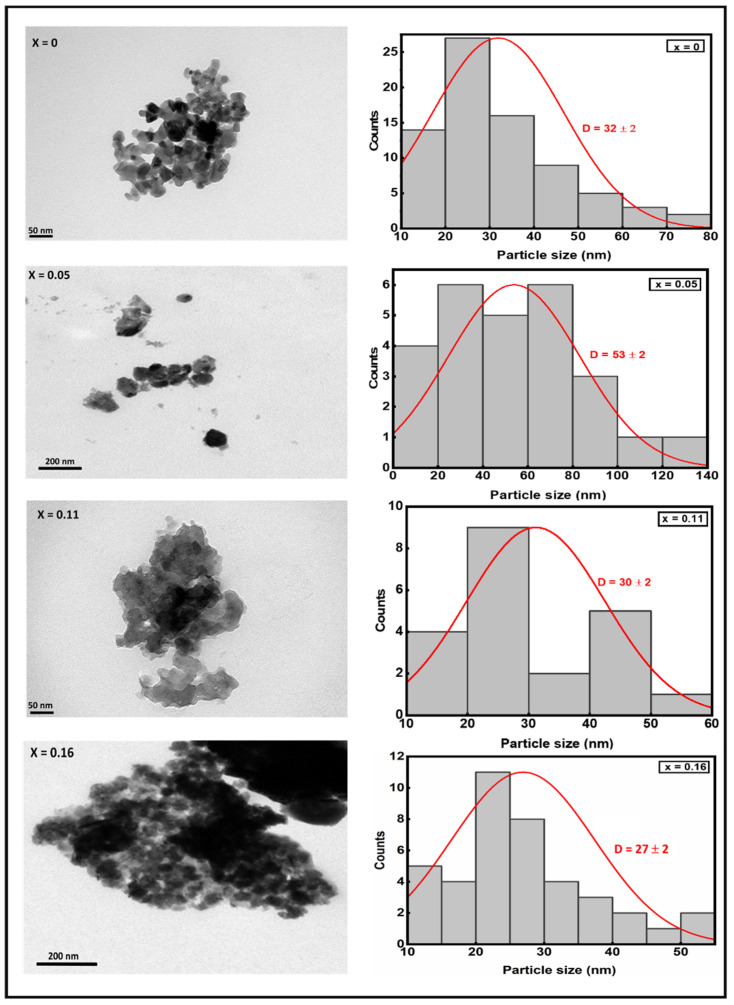
TEM images and particle size distribution of CuIn_1-x_Zn_x_Se_2_ powders.

**Figure 8 materials-15-01436-f008:**
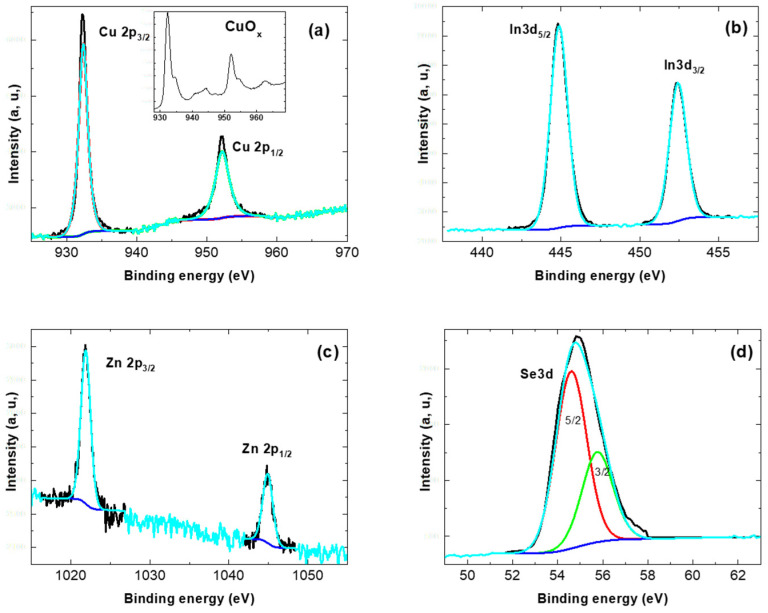
XPS spectra of CuIn_1-x_Zn_x_Se_2_ powder with x = 0.21, (**a**) Cu2p, (**b**) In3d, (**c**) Zn2p and (**d**) Se3d.

**Figure 9 materials-15-01436-f009:**
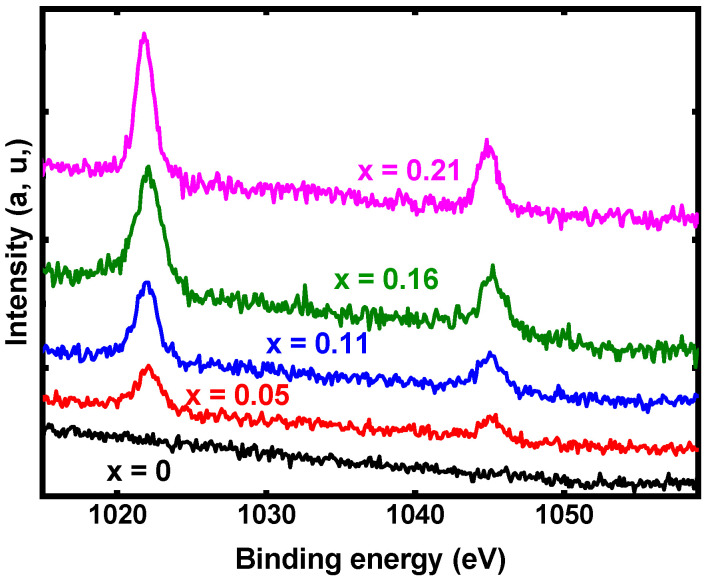
Evolution of the XPS spectrum of Zn2p with the Zn concentration in CuIn_1-x_Zn_x_Se_2_ powder.

**Figure 10 materials-15-01436-f010:**
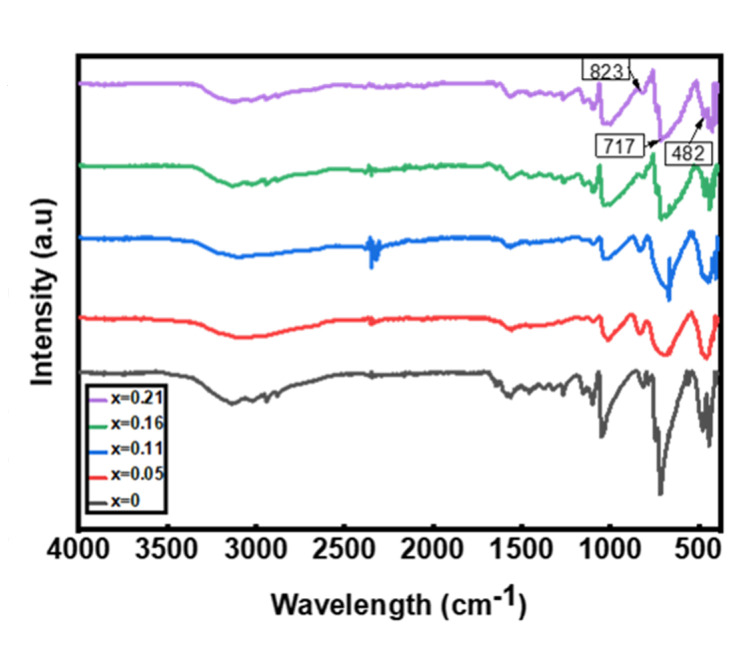
FT-IR spectra of CuIn_1-x_Zn_x_Se_2_ powders.

**Figure 11 materials-15-01436-f011:**
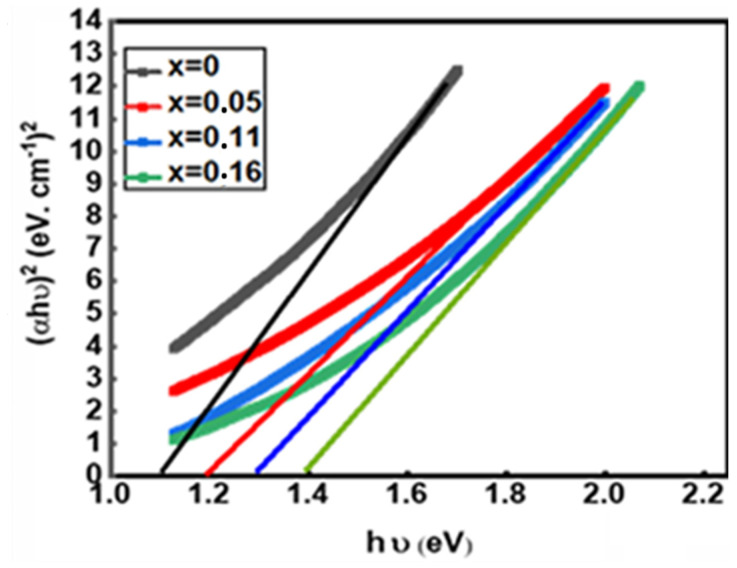
Tauc function of CuIn_1-x_Zn_x_Se_2_ powder.

**Figure 12 materials-15-01436-f012:**
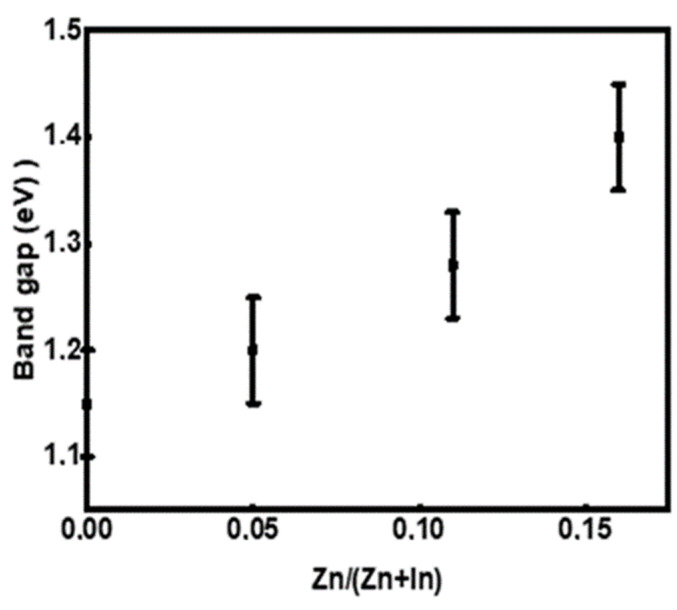
Band gap as a function of x of CuIn_1-x_Zn_x_Se_2_ powders.

**Table 1 materials-15-01436-t001:** Chemical composition and structural parameters of CuIn_1-x_Zn_x_Se_2_ (space group $1\bar{4}2d$).

Sample Name	Zn/(In+Zn)	2θ	FWHM	D (nm)	a (Å)	c (Å)
x = 0	0	26.69	0.3226	25	5.782	11.668
x = 0.05	0.05	26.67	0.2322	35	5.784	11.571
x = 0.11	0.11	26.65	0.2715	30	5.785	11.593
x = 0.16	0.16	26.71	0.3831	21	5.779	11.540
x = 0.21	0.21	26.73	0.4837	17	5.777	11.523

**Table 2 materials-15-01436-t002:** Composition of CuIn_1-x_Zn_x_Se_2_ powders as a function of the composition of the precursor solution.

x	Precursor Composition				Measured by EDX			
Zn/ (In + Zn)	at % Cu	at % In	at % Zn	at % Se	at % Cu	at % In	at % Zn	at % Se
0	25	25	-	50	25 ± 1	23± 1	-	52 ± 1
0.05	25	23.5	1.5	50	24 ±1	18± 1	1 ± 1	57 ± 1
0.11	25	22.5	2.5	50	25±1	17± 1	2 ± 1	56 ± 1
0.16	25	21.5	3.5	50	31±1	16± 1	3 ± 1	50 ± 1
0.21	25	20.5	4.5	50	29±1	15± 1	4 ± 1	52 ± 1

## Data Availability

Data sharing is not applicable to this article.

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
