# Peer review of "Syntheses and Characterizations of CuIn1-xZnxSe2 Chalcopyrite Nanoparticles"

_materials, 2022, doi:10.3390/ma15041436_

Round 1

Reviewer 1 Report

Manuscript from Khedidja Benameur deals with Syntheses and Characterizations of CuIn1-xZnxSe2 Chalcopyrite Nanoparticles. It is worth to mention that I reviewed also previous version of the manuscript.

I really appreciate changes you made in the manuscript and I think manuscript quality is high enough in order to be published.

On the other hand, my strongest negative point deals with issue how the particle size is changing with increasing Zn content.

line 211-213: I am not convinced that "These results show that the particle size decreases with the percentage increase of Zn in CuInZnSe powders." How do you measure nm dimensions in micrometer resolution of SEM? Moreover, particle size was measured by XRD and I see firstly decrease and than increase of particle size with x! Be more careful!

Maybe you should define what you mean by "particle size" in SEM image. Is it diameter of ball micro-sphere? Or size of nano-plate? How do you measure this (e.g. 53 - 62 nm at line 207) in micrometer resolution?

line 226-227 and 291-292 again: I am not sure where your conclusion that "particle size decreases with the percentage increase of Zn" comes from. Although even gaussian fit of histograms in fig. 7 can be questionable I do not see such straightforward conclusion from fig. 7. Even not-monotonic dependence can occur.

Moreover, some minor comments:

Line 4: I recommend to delete "." after the surnames of authors

Line 61: UV-visible absorbance

line 75: 180 °C

line 105: cm-1 (upper index)

line 167: polycrystalline

line 167: explanation of what? number of dislocations? Incorporation of Zn into crystal structure?

Figure 4: I recommend to change color of decription of right y-axis (delta) to red

Can you insert inset into fig. 8 with the range from 1000 cm-1? Will it help for claim in lines 244-247?

line 256: absolute ethanol? Why "absolute"?

line 267: with 0.05 eV estimated error - delete estimated

I recommend to restrict upper limit of y-scale in fig. 10 to 1.5

lines 296 add spaces "CuIn1-xZnxSe2nanoparticlescanbeused"

Check also missing spaces in references (lines from 311)

I hope my comments will help you to improve the manuscript and publish it.

Author Response

February 08, 2022

Ms. Ref. No.:

Title: Syntheses and Characterizations of CuIn1-xZnxSe2 Chalcopyrite Nanoparticles

Replies to the reviewer comments

Dear Pr Belle Sang, General Editor,

Thank you for giving us the opportunity to submit a revised draft of our manuscript “Syntheses and Characterizations of CuIn1-xZnxSe2 Chalcopyrite Nanoparticles.” for publication in Materials. We appreciate the time and effort that you and the reviewers dedicated for providing feedback on our manuscript and are grateful for the insightful comments and valuable improvements to our paper. We have incorporated all suggestions made by the reviewers. Those changes are highlighted within the manuscript. Please see below, for a point-by-point response to the reviewers’ comments and concerns.

Response to the reviewer’s comments.

Reviewer #1

On the other hand, my strongest negative point deals with issue how the particle size is changing with increasing Zn content. line 211-213: I am not convinced that "These results show that the particle size decreases with the percentage increase of Zn in CuInZnSe powders." How do you measure nm dimensions in micrometer resolution of SEM? Moreover, particle size was measured by XRD and I see firstly decrease and than increase of particle size with x! Be more careful!

Response: 

The figure 4 shows the variation of the grain size of the CuInZnSe powders as a function of the atomic ratio Zn/(Zn + In). The grain size or the crystalline quality first increases indicating the zinc atoms occupy vacant indium size. Then it decreases at the threshold of 0.05 with the increase of the atomic ratio Zn/(Zn + In). The zinc atoms are substituted to indium atoms. The decrease of the crystallites size is probably due to the small ion size of Zn2+ (radius = 0.74 Å) compares to the one of In3+ (radius = 0.81 Å). We have modified the manuscript as follows:

We note that in the figure 4 showing the variation of the grain size of the CuInZnSe powders as a function of the atomic ratio Zn/(Zn + In) that the grain size or the crystalline quality first increases indicating the zinc atoms occupy vacant indium size. Then it decreases at the threshold of 0.05 with the increase of the atomic ratio Zn/(Zn + In). The zinc atoms are substituted to indium atoms. The decrease of the crystallites size is probably due to the small ion size of Zn2+ (radius = 0.74 Å) compared to the one of In3+ (radius = 0.81 Å).

May be you should define what you mean by "particle size" in SEM image. Is it diameter of ballmicro-sphere? Or size of nano-plate? How do you measure this (e.g. 53 - 62 nm at line207) in micrometer resolution?  

Response: Particle size" in SEM image. It is diameter of ballmicro-sphere software. We have modified the manuscript as follows:

The SEM images in the figure 6 show that the CuIn1-xZnxSe2 powders include ball microspheres and interconnected nano-platelets. These nano-objects look like desert rose-like structures at a higher magnification [18, 19] as shown in the figure 5.The size of the nanoballs is larger than the average crystallite size determined by XRD. This can be explained by the fact that each particle observed by SEM is formed of several crystallized grains. Besides, we do not measure the same entity; XRD gives the size of the grains perpendicularly while the SEM gives the size of the grains on the surface so parallel to the surface. These results show that the diameter of ball micro-sphere decreases with the percentage increase of Zn in CuInZnSe powders.

line 226-227 and 291-292 again: I am not sure where your conclusion that "particle size decreases with the percentage increase of Zn" comes from. Although even gaussian fit of histograms in fig. 7 can be questionable I do not see such straight forward conclusion from fig. 7. Even not-monotonic dependence can occur. 

Response: Thank you for your comment. "The particle size decreases with the percentage increase of Zn in CuInZnSe powders." has been removed in conclusion. We have modified the manuscript as follows:

The TEM images in figure 7 show that the powders consist of spherical shaped agglomerated nanoparticles. The morphology of the particles not being perfectly spherical also contributed towards deviation from mean size shape in which many small building blocks are connected in a large network. The size of these particles was determined by statistical counting using the Imagej software, which ranges from 55 to 25 nm. These results show that we can think that they give a correct idea since they are confirmed by the DRX.

Moreover, some minor comments:

Line 4: I recommend to delete "." after the surnames of authors.

 Response:  The dots have been deleted after the names

Line 61: UV-visible absorbance, line 75: 180 °C, line 105: cm (upper index), line 167: polycrystalline.

Response: The corrections in the text have been done.

line 167: explanation of what? number of dislocations Incorporation of Zn in to crystal structure?

Response: Thank you for your comment. We have modified the manuscript as follows:

It has become a fundamental building block in the explanation of dislocations Incorporation of Zn in to crystal structure.

Figure 4: I recommend to change color of decription of right y-axis (delta) to red

Response: We have modified the color of the right axis of the figure of the manuscript as follows:

Figure 4. Variation of the dislocation density (δ) and grain sizes (D) for CuIn1-xZnxSe2 powders as a function of the atomic ratio Zn/(In+Zn)).

Can you insert inset into fig. 8 with the range from 1000 cm-1? Will it help for claim in lines 244-247.

Response: We did the XPS analysis and changed the range of the study according to the lines 244-247. We have modified the manuscript as follows:

            Concerning XPS measurements, an Axis Nova instrument from Kratos Analytical spectrometer with Al Kα line (1486.6 eV) as excitation source has been used. The core level spectra were acquired with an energy step of 0.1 eV and using a constant pass energy mode of 20 eV, (energy resolution of 0.48 eV). Concerning the calibration, binding energy for the C1s hydrocarbon peak was set at 285 eV. Then data were analysed with the CasaXPS software.

4.3 Elemental analysis of CuIn1-xZnxSe2 nanopowders

In Figure 8, the shape of the spectra of the different elements, Cu2p3/2, In3d5/2, Zn2p3/2 and Se3d present in the samples, demonstrate that they are not oxidized. This is particularly explicit in the case of Cu, of which it is known that the spectrum of copper oxide is completely different from that of Cu present in chalcopyrite compounds (Inset Fig. 8a) [22, 23].The binding energies of the different elements can be deduced from Figure x. For Cu 2p3/2 and 2p1/2they are 932.4 eV and 952.2 eV respectively. From Figure 8b the In3d5/2 and In 3d3/2 binding energies are 444.9 eV and 452.4 eV respectively, while the values estimated for Se3d5/2 and Se3d3/2 are 54.5 eV and 55.7 eV. These binding energies are in agreement with those reported in the literature for chalcopyrite structures [24, 25] Beside, Fig. 8c shows the XPS spectrum of Zn2p, where it can be seen that Zn2p3/2 and Zn2p1/2 are situated at 1021.9 eV and 1044.9eV respectively. It means that there is a small increase of the Zn orbital energies in comparison with the expected values for pure Zn [22].This small shift corresponds to electrons exchange with Se, which confirms the Zn substitution to In.

Figure 8. XPS spectra of CuIn1-xZnxSe2 powder with x = 0.21, (a) Cu2p, (b) In3d, (c) Zn2p and (d) Se3d.

In Figure 9, it can be seen clearly that the intensity of the Zn signal increases with the x value. The x values were determined by EDS and it is remarkable that the relative intensity of the lines detected by XPS follows these values fairly closely. Thus, the XPS study confirms the presence of Zn and its interaction with the atoms of the CuIn1-xZnxSe2 compound.

Figure 9. Evolution of the XPS spectrum of Zn2p with the Zn concentration in CuIn1-xZnxSe2 powder.

line 256: absolute ethanol? Why "absolute"?

Response: Absolute ethanol is a purer form of ethanol than any other form. This is because it contains 99-100% ethanol. We have modified the manuscript as follows:

 absolute ethanol (≥ 99.9%)

line 267: with 0.05 eV estimated error – delete estimated

Response: We have deleted this estimated error.

I recommend to restrict upper limit of y-scale in fig. 10 to 1.5

Response: We have did it. We have modified the manuscript as follows:

Figure 10. Band gap as a function of x of CuIn1-xZnxSe2 powders.

lines 296 add spaces "CuIn1-xZnxSe2 nanoparticles can be used". 

Responses: We have did it.

Check al so missing spaces in references (lines from 311). We have did it.

Responses: The missing spaces are checked and corrected in the references

Reviewer 2 Report

The objective of this work is to study the effect of the concentration of Zn atoms on the structural, morphological and optical properties of CuIn1-xZnxSe2 nanoparticles.The authors prepared five samples of CuIn1-xZnxSe2 (X=0, 0.05, 0.11, 0.16 and 0.21) by the solvothermal method.The samples are analyzed by six techniques: XRD, EDX, SEM, MET, FTIR and UV-Vis. The manuscript presents an interesting study. However, there are still some issues need to be well addressed before publication. The detailed comments are listed below:

In the experimental part (lines 67, 72 and 89), please check the precursor Cul or CuCl2.

In line 115: In the DRX diagrams, the peak (112) is the most intense. This does not mean that the powder has an orientation along the (112) direction.

Lines 137 and 138

In table 1, the column of the position of the peak (112) must be placed before that of FWHM.

The position of the peak is expressed in degrees. Please let us know how you calculated the absolute uncertainties on a, c and grain size.

In figure 3, the shift of the position of the peak (112) towards the highest angles is not clear, especially for the samples x=0.05 and x=0.11. It is necessary to normalize to the value 1 the intensities of the peaks in this region (25°-28°).

Line 164: δ represents the dislocation length per unit of surface and not per unit of volume.

Line 195: In table 2, how do you explain the slight difference between the concentration of Zn in the starting precursors and in the powders?

Line 210: The values of the atomic concentrations of the sample x=0.05 reported on the EDX spectrum (figure 5) are not the same reported in table 2.

Line 210 Fig 5: There is a repetition of “of CuIn1-xZnxSe2 powders”

Line 258: It must be indicated that CuInZnSe2 is a direct gap semiconductor in order to be able to extract Eg from curve 9.

To calculate the absorption coefficient a from the transmittance (absorbance) curve you need the thickness.

Please add the method used to estimate the thickness of your samples.

Line 274 in Fig 10: The word “Zn(In+Zn)” should be replaced by “Zn/(In + Zn)”

References :  There is an overlap in the references. You must write correctly.

Author Response

Ms. Ref. No.:

Title: Syntheses and Characterizations of CuIn1-xZnxSe2 Chalcopyrite Nanoparticles

Replies to the reviewer comments

Dear Pr Belle Sang, General Editor,

Thank you for giving us the opportunity to submit a revised draft of our manuscript “Syntheses and Characterizations of CuIn1-xZnxSe2 Chalcopyrite Nanoparticles.” for publication in Materials. We appreciate the time and effort that you and the reviewers dedicated for providing feedback on our manuscript and are grateful for the insightful comments and valuable improvements to our paper. We have incorporated all suggestions made by the reviewers. Those changes are highlighted within the manuscript. Please see below, for a point-by-point response to the reviewers’ comments and concerns.

Response to the reviewer’s comments.

Reviewer #2

In the experimental part (lines 67, 72 and 89), please check the precursor Cul or CuCl2.

Response: The manuscript was checked and the correct precursor is CuCl

In line 115: In the DRX diagrams, the peak (112) is the most intense. This does not mean that the powder has an orientation along the (112) direction.

Response: Thank you for your comment. « The crystallization of the powders is oriented 112 towards the main line (112) which is the most intense” has been removed.

Lines 137 and 138 In table 1, the column of the position of the peak (112) must be placed before that of FWHM.

Response: We have modified the manuscript as follows:

Table 1. Chemical composition and structural parameters of CuIn1-xZnxSe2 (space group)

Sample name

Zn/(In+Zn)

2

FWHM

D (nm)

a (Å)

c (Å)

x=0

0

26.69

0.3226

25

5.782

11.668

x=0.05

0.05

26.67

0.2322

35

5.784

11.571

x=0.11

0.11

26.65

0.2715

30

5.785

11.593

x=0.16

0.16

26.71

0.3831

21

5.779

11.540

x=0.21

0.21

26.73

0.4837

17

5.777

11.523

The position of the peak is expressed in degrees. Please let us know how do you calculated the absolute uncertainties on a, c and grain size.

ResponseThank you for your comment, after verification the table 1, we discussed this appropriately and we are really sorry that we relied on a method, and after the discussion, we found out that it was wrong, and as you said, the angle is in degrees that it is not possible to calculate absolute uncertainties. We corrected the error by removing uncertainty from Table1. We have modified the manuscript as follows:

Table 1. Chemical composition and structural parameters of CuIn1-xZnxSe2 (space group)

Sample name

Zn/(In+Zn)

2

FWHM

D (nm)

a (Å)

c (Å)

x=0

0

26.69

0.3226

25

5.782

11.668

x=0.05

0.05

26.67

0.2322

35

5.784

11.571

x=0.11

0.11

26.65

0.2715

30

5.785

11.593

x=0.16

0.16

26.71

0.3831

21

5.779

11.540

x=0.21

0.21

26.73

0.4837

17

5.777

11.523

Figure 4. Variation of the dislocation density (δ) and grain sizes (D) for CuIn1-xZnxSe2 powders as a function of the atomic ratio Zn/(In+Zn)).

In figure 3, the shift of the position of the peak (112) towards the highest angles is not clear, especially for the samples x = 0.05 and x = 0.11. It is necessary to normalize to the value 1 the intensities of the peaks in this region (25°-28°).

Response: we normalized to the value 1 the intensities of the peaks in this region (25°-28°).

Figure 3. Zoom of XRD patterns for CuIn1-xZnxSe2powders.

Line 164: δ represents the dislocation length per unit of surface and not per unit of volume.

Response: We corrected this error. We have modified the manuscript as follows:

The dislocation density (δ) may also be considered as a measure of the defects amount in the compound. δ is defined as the length of the dislocation lines per unit surface of unit cell of the crystal was calculated using the Williamson and Smallman formula:

Line 195: In table 2, how do you explain the slight difference between the concentration of Zn in the starting precursors and in the powders? 

Response: If we take into account, the uncertainty there is no real difference. Moreover, the measurement gives rather a trend, namely that, as expected, the at.% Zn of the final product increases with the initial concentration, we have modified the manuscript as follows:

A slight variation was also noticed between the composition of the individual particles measured by EDX and the ratio of the precursor taken at the starting precursor composition. If we take into account, the uncertainty there is no significative difference. Moreover, the measurement gives rather a trend, namely that, as expected, theat.% Zn of the final product increases with the initial concentration.

Line 210: The values of the atomic concentrations of the sample x=0.05 reported on the EDX spectrum (figure 5) are not the same reported in table 2.

Response: In the table, we have given the average value of 10 samples, but the spectrum gives the value of only one sample as an example.

Table 2. Composition of CuIn1-xZnxSe2 powders as a function of the composition of the precursor solution.

x

Precursor composition

Measured by EDX

Zn/(In+Zn)

at.% Cu

at.% In

at.%

 Zn

at.% Se

at.% Cu

at.% In

at.% Zn

at.% Se

0

25

25

-

50

25

23

-

52

0.05

25

23.5

1.5

50

24

18

    1

57

0.11

25

22.5

2.5

50

25

17

2

56

0.16

25

21.5

3.5

50

31

    16

3

50

0.21

25

20.5

4.5

50

29

15

4

52

To calculate the uncertainty, we used this formula:

Example for the sample x = 0.05

1

2

3

4

5

6

7

8

9

10

Cu

23.74

23.06

23.65

24.67

24.00

23.82

23.90

24.40

24.60

23.20

In

17.95

18.83

17.65

21.21

18.10

17

17.02

18

16.20

18.04

Zn

1.11

1.14

1.11

1.12

1

1.02

1.03

0.9

1.01

1.04

Se

57.2

56.97

57.59

53

56.9

58.12

58.05

56.7

58.19

57.72

We have modified the manuscript as follows:

Figure 5. SEM images of CuIn1-xZnxSe2 powders.

Line 210 Fig. 5: There is a repetition of “of CuIn1-xZnxSe2 powders”

ResponseThank you for your remark, we have modified the manuscript as follows:

Figure 5. SEM images of CuIn1-xZnxSe2 powders.

Line 258: It must be indicated that CuInZnSe2 is a direct gap semiconductor in order to be able to extract Eg from curve 9.

Response: We have added this comment in the manuscript:

Chalcopyrite CuIn1-xZnxSe2 is a direct bandgap semiconductor, so to determine their bandgap , we extrapolate the linear part of the curve representing the (αhv)2 function to zero. The figure 9 shows the result of (αhν)² as a function of hν. Based on these results the energy of the band gap can be estimated with an estimated error of 0.05 eV. The figure 10 represents the variation of the values of the forbidden band as a function of the atomic ratio Zn/(In + Zn). From these results we deduce that the increase of the atomic ratio Zn/(In + Zn) is followed by the increase of the band gap and this may be due to the displacement of the CuIn1-xZnxSe2 conduction band to higher energy levels and the valence band to lower energy levels. Therefore, the band gap of these compounds increases [24].

To calculate the absorption coefficient a from the transmittance (absorbance) curve you need the thickness. Please add the method used to estimate the thickness of your samples.

Response: Here is a link that shows this method:

https://www.youtube.com/watch?v=tbyJjCMkDtc

The following equation,

(ahn)n = K(hn-Eg)

is know as Tauc and Dvid-Mott relation. This relation is used to probe the optical band gap energy of nanoparticles from UV-Vis absorption spectroscopy. In this equation, a is the absorption coefficient, hn is the incident photon energy, K is the energy independent constant and Eg in the optical band gap energy of nano material. In this equation, the exponent ‘n’ represent the nature of transition. For direct band gap material n = 2 while for indirect n = 1/2.

In the Tauc plot method, we plot the energy on x-axis while (ahn)n on y-axis. Then we draw the tangent line on the curve where a = 0. The point where it touches the x-axis is the optical band gap energy of the material. In the UV-Vis Spectroscopy, the data, which is given to us, have wavelength and absorbance.

Therefore, we have to convert wavelength to energy and calculate the absorbance coefficient. ‘a’ from the absorbance data.

In the Tauc plot, we have (ahn)non y-axis here hnis the incident photonic energy and ‘a’ is the absorbance coefficient, we can calculate ‘a’ from absorbance data using Beer lambert’s law.

Here in the figure, ‘I’ is the intensity of transmitted light, ‘Io’ is the intensity of incident light, ‘a’ is the absorbance coefficient and ‘L’ is the Path-length of light in which absorbance take place.

……(1)

We can modified eq (1) as,

Taking log on both sides.

 (powerrule of log)

Multiply ‘–‘ on both sides.

……(2)

As

Absorbance = A =

(please note, absorbance is dimensionless quantity)

We can re-write eq (2) as,

            where, log(e) = 0.4343

So

……(3)

Now we consider a standard cuvette which is commonly used in spectrophotometers,

This is a standard Quartz cuvette.,

In standard cuvettes the path is usually equal to 10 mm, i.e

L = 10 mm = 1 cm

Therefore, the equation (3) can be written as,

……(4)

The unit of absorbance coefficient would be cm-1, because 2.303 is a constant and Absorbance is dimensionless quantity.

Now we can combine eq (4) with energy to get our desire Tauc relation i,e

(ahn)n=(Absorbance coefficient x Energy)n

The unit of this whole things would be (eVcm-1). This the exact equation, which I have used in my manuscript*How to calculate optical band gap energy from absorption data using Tauc plot method*

we have modified the manuscript as follows:

In Tauc Plot method, it is necessary to extrapolate to zero the linear part of the curve representing Davis and Mott relation (αhν)n = K(hν-Eg) [29],where α is absorption coefficient, K is a constant, hν the photonic energy, Eg the energy of the band gap and n represent the nature of transition. For direct band gap material n = 2 while for indirect n = 1/2. Davis and Mott relation can be further expressed as (2.303*A*1240/l)n = K(1240/l - Eg) [30] , Where A and λ are absorbance and wavelength respectively obtained from the absorption spectra of the nanoparticles. A plot this relation gives an absorption curve in which its tangent gives the energy band gap of the nanoparticles [30].

Line 274 in Fig10: The word “Zn(In+Zn)” shouldbereplaced by“Zn/(In + Zn)”

Response: We corrected this error. We have modified the manuscript as follows:

Figure 10. Band gap as a function of x of CuIn1-xZnxSe2 powders.

References: There is an overlap in the references. You must write correctly.

Response: Thank you for your remark. We corrected all the errors in the references.
